# Investigating a Prognostic Factor for Canine Hepatocellular Carcinoma: Analysis of Different Histological Grading Systems and the Role of PIVKA-II

**DOI:** 10.3390/vetsci9120689

**Published:** 2022-12-10

**Authors:** Lorella Maniscalco, Katia Varello, Emanuela Morello, Vittoria Montemurro, Matteo Olimpo, Davide Giacobino, Giuseppina Abbamonte, Cecilia Gola, Selina Iussich, Elena Bozzetta

**Affiliations:** 1Dipartimento Scienze Veterinarie, Università degli Studi di Torino, 10045 Grugliasco, Italy; 2Istituto Zooprofilattico Sperimentale del Piemonte, Liguria e Valle d’Aosta, 10154 Torino, Italy

**Keywords:** PIVKA-II, canine hepatocellular carcinoma, prognostic factors

## Abstract

**Simple Summary:**

PIVKA-II is an aberrant form of vitamin K that is increased in human coagulation disfunctions and in some neoplastic diseases. In veterinary medicine, PIVKA-II concentrations can be useful to identify patients with coagulative disorders in plasma and tissues, but its role as marker for hepatocellular carcinoma has not been investigated previously. In this study we characterized ed the expression of PIVKA-II in canine hepatocellular carcinomas in relation with the prognosis and some histological grading systems with the aim of finding useful prognostic factors for this canine tumour.

**Abstract:**

Background: Hepatocellular carcinoma (HCC) in dogs is uncommon and often associated with a good prognosis, although some cases prove to be aggressive. In human oncology HCC is often very aggressive and diagnostic methods and prognostic factors are widely used to predict its biological behaviour. These include the expression of PIVKA-II. Methods: in order to identify a prognostic factor for canine HCC, we applied different methods of histological grading and investigated PIVKA-II expression in 22 HCC of dogs treated surgically and followed clinically for at least 2 years. Results: Nineteen patients analysed have passed the observation period without tumour recurrence, while 3 died following the development of metastases. PIVKA-II was positive in 15/22 cases without correlation with prognosis or tumoural grading even if a trend of PIVKA-II negativity in low WHO grades as well as increased number of PIVKA-II positive cases in higher WHO grades weres observed. Conclusions: This work showed that, PIVKA-II cannot be considered either as a marker of malignancy or as a prognostic marker for canine HCC. The poor prognosis depends usually on the clinical presentation. Thus prognostic parameters in canine HCC able to predict its aggressive behaviour through histological examination are still missing. The most promising method, limited to our study, seems to be the WHO histological grading.

## 1. Introduction

Primary liver tumours represent less than 1.5% of all canine neoplasia [1]. Malignant tumours of the liver include hepatocellular carcinoma (HCC), cholangiocellular carcinoma, neuroendocrine tumours, and sarcomas. Hepatocellular carcinoma is the most common, accounting for approximately 50–77% of primary hepatobiliary tumours in dogs [2]. Hepatocellular carcinoma is an epithelial tumour primarily arising from hepatocytes and is usually found in older dogs. No sex predisposition or specific risk factors have been recognized, although male dogs may be overrepresented [2,3]. 

Currently, canine HCC is not considered as a potential model in comparative oncology. Unlike human hepatocellular carcinoma, the canine counterpart is much less frequent, probably because not associated with lifestyle and often not as aggressive if the massive presentation is considered [2]. Moreover, a grading method for hepatocellular carcinomas is not commonly applied in veterinary oncology, unlike in human where, according to the literature, several methods are described [4].

Protein induced by vitamin K absence or antagonist II (PIVKA-II) is a substance produced under conditions of vitamin K deficiency or in the presence of vitamin K antagonists such as warfarin. It was first described in 1984 by Liebman et al. as potential tumour marker for HCC [5]. In later years many reports have been published about the clinical usefulness of serum PIVKA-II and an association with the prognosis of HCC has been suggested in human oncology [6,7,8]. 

PIVKA-II is used as retroactive indicator of vitamin K level. [9]; this aspect has been deeply studied in human medicine so that high PIVKA-II serum concentrations have been demonstrated to be indicator of coagulative pathologies and some neoplastic diseases such as in HCC [10,11,12]. Some authors investigated PIVKA-II expression by immunohistochemistry in human HCC and Miskad and colleagues demonstrated that PIVKA-II was expressed even in small-sized or well-differentiated neoplastic cells, but not in adenomatous hyperplasia, assessing that PIVKA-II is a useful immunohistochemical marker for HCC [13]. Moreover, the potential relationship between PIVKA-II expression and the different grades of HCC was investigated through immunohistochemical studies of paraffin-embedded tissues [14]. 

We previously investigated the expression of PIVKA-II in canine liver and kidney proving that PIVKA-II can be an indicator of coagulopathies or ingestion of anticoagulants compounds [15].

Whereas the immuno-expression of PIVKA-II in tissues is evaluated for its importance as a prognostic factor in human patients with cancer, it has not yet been studied in animals with this purpose. Therefore, the aim of the present study is to evaluate the expression of PIVKA-II in canine hepatocellular carcinoma and to investigate its potential prognostic role in relation to clinical follow-up.. In addition, since a canine HCC histological grade is not currently applied, an attempt was made to use a histologic grade method currently used to evaluate its applicability as a predictor of prognosis and its positivity in correlation with PIVKA expression.

## 2. Materials and Methods

### 2.1. Sample Collection and Clinical Follow-Up

Tissue samples were examined from 22 cases of spontaneous hepatocellular carcinomas surgically treated between 2005 and 2021 at the Department of Veterinary Sciences of the University of Turin. None of the dogs included in this study had evidence of macroscopic metastases at presentation based on the imaging diagnostics used to stage patients (CT total body or abdominal ultrasound plus three views thoracic radiographs). All dogs included in this study were surgically treated for complete removal of the liver tumour.

Clinical examination, abdominal ultrasound and thoracic radiographs were performed every 3 months in the first year after surgery and then every 6 months in the second year. Long term follow up was obtained by phone interview to the owners. If still alive after 2 years from surgery the dogs were censored for statistic purpose. For animals that died for tumour-related causes within the 2-year period, overall survival (OS) was considered as the days between the surgery and death, while the disease-free interval (DFI) was considered as the number of days between surgery and tumour recurrence and/or evidence of metastatic disease.

### 2.2. Histological Examination

Portions of neoplastic tissue were fixed in 4% neutral buffered formalin, paraffin embedded, sectioned at 4 μm, and stained routinely with hematoxylin and eosin (HE). Histological grading related to nuclear pleomorphism, nucleolar pleomorphism and architecture was applied by two different pathologists separately (L.M. and K.V.) according to histological parameters suggested by both WHO [16] and Martins-Filho et al (2017) histological gradings (Table 1) [4].

### 2.3. Immunohistochemistry

Immunohistochemical staining against PIVKA-II was performed using the previously described technique [15] using primary polyclonal antibody, anti-PIVKA-II (Rabbit polyclonal antibody; 1:2000 dilution, synthesized by commission by Diatheva Srl, Cartoceto, Italy).. Instead of the primary antibody, the immunoglobulin fraction of mouse non-immune serum was used to incubate the negative control sections. Two impartial observers who were not informed of the clinical and histological diagnosis viewed and evaluated the immunostained sections under a light microscope. (L.M., K.V.) 

### 2.4. Statistical Analysis

IHC results and clinicopathologic outcomes were grouped into contingency tables and analyzed using Fisher’s exact test or χ^2^ test. The analysis of OS and DFI was performed using the Kaplan Meier method with a log rank test. Data were computed with MedCalc Statistical Software v.13.3 (MedCalc Software bvba, Ostend, Belgium).

## 3. Results

### 3.1. Clinico-Pathological Results

Twenty-two dogs were enrolled in the present study. Females and males were equally represented. Results are summarized in Table 2. Most of the patients included in the present study (17/22; 77.27%) passed the observation period without developing relapses. Two dogs (13.64%) died as a result of surgical complications a few days after surgery, while only three dogs developed metastases and died for tumor related causes. 

The application of histological grades identified an extreme variability in terms of grade when the method proposed by Martins-Filho and colleagues was applied [4], depending on whether the grade of architecture, nuclear or nucleolar morphology was assessed. The results are summarized in Table 2. PIVKA-II immunopositivity was observed in the cytoplasm. However, a weak positivity was present in the peritumoural liver tissue in all cases (Figure 1). Fifteen out of 22 cases (68.18%) were PIVKA-II positive while 7/22 were negative (31.82%).

### 3.2. Statistical Analysis 

Statistical analyses comparing PIVKA-II expression and clinico-pathological findings show no statistical associations; regarding the different applied histological gradingsa statistical association between WHO grade 2 and PIVKA-II positivity (*p* < 0.05) together with an increase of PIVKA-II negative cases in low WHO grades as well as increased number of PIVKA-II positive cases in higher WHO grades (Figure 2) were observed. 

Moreover analyzing the others histological grades a statistical association between nucleolar pleomorphism grade 4 and positivity to PIVKA-II (Fisher’s Exact test *p* < 0.05) was found (Figure 3).

Long rank tests performed to compare the IHC results and the OS and DFI times showed no statistical correlation. In addition, analyses aimed at evaluating possible association between the different histological grades applied and survival showed a significant association between grade 3 nuclear pleomorphism and reduced survival, a finding that is not confirmed in grade 4 cases. Other statistically significant associations regarding survival and the parameters analyzed were not found. 

## 4. Discussion

To the awareness of the Authors PIVKA-II immune-expression in canine HCC was not previously studied. PIVKA-II is an abnormal form of prothrombin that is induced by hepaticcells in case of vitamin k deficiency, such as in coagulative disorders, in presence of vitamin K antagonist [17,18] or in some neoplastic diseases beig used as prognostic marker in human HCC since [19,20,21]. This aberrant prothrombin is used as an indicator of coagulative disorders also in canine specie as PIVKA-II test on plasma samples was shown by Mount and colleagues to be diagnostically helpful for differentiating anticoagulant poisoning from other coagulopathies in dogs [22]. We previously demonstrated that the expression PIVKA-II can be expressed in hepatic and renal tissue of dogs died for anticoagulant poisoning but also in subjects died for coagulative disorders or having hepatic degeneration [15]. In the current study we found that PIVKA-II is heteregenous in HCC but it is always positive, even though weakly, in the normal hepatic tissue surrounding tumour. This result is in line with our previous study in which we observed that degenerated hepatocytes expressed PIVKA-II probably due to liver dysfunction induced coagulative disorders [15]. It is plausible to hypothesize that under conditions of neoplasm compression, the liver, which histologically appears degenerated, also has altered metabolism of coagulation parameters resulting in the expression of PIVKA-II, but we cannot support this result with the altered serum coagulation parameters in our study, both due to the absence of such clinical data and because it is conceivable that since the dogs underwent surgery, their coagulation parameters were not too much altered compared with normal parameters. This finding seems to differ from that of some studies on human hepatic carcinoma tissues, where PIVKA-II appeared to be a helpful indicator of malignancy by being significantly positive in neoplastic cells and negative in surrounding cirrhotic or regenerative hepatic tissue. [13,23]. Since PIVKA-II did not prove to be a good marker for discriminating neoplastic changes in the liver in dogs, we investigated whether the expression of PIVKA-II could be different in relation to the histological differentiation of HCC. Since HCC in the dog normally does not have an aggressive biological behavior, there is not a useful histological grading system that is applied in veterinary pathology and therefore we used different methods proposed in human pathology, in human pathology considering WHO grading, architecture, nuclear and nucleolar pleomorphism.

As shown in Table 1, there is no correspondence between the grades associated with the different parameters and extreme variability is observed, as in cases n 4 and 7 where the parameters related to nuclear and nucleolar morphology result in a low grade, while the architecture of the neoplasm corresponds to a high grade. Limited to the few cases analysed, the most applicable method in veterinary medicine would seem to be the WHO method as described by Martins-Filho and colleagues regarding human HCC. Although we did not observe a statistically significant association between the histological grade and the survival time of our patients, we cannot rule out the possibility that in a larger number of cases this method may prove useful, both because in our study many cases were censored being lost after the observation period and due to tha fact that few cases that showed metastases had a WHO grade of 3 or 4. 

## 5. Conclusions

In the present study, limited to the analyzed cohort of cases, PIVKA-II does not appear to be useful either as a diagnostic or prognostic marker in canine HCC differently to what has been observed in human HCC. This finding seems to be different to needs to be confirmed in a larger cohort of cases. The need remains to find reliable prognostic parameters in canine HCC that can predict its aggressive behaviour, which, although rarely observed, is present in some cases. The most promising method, limited to our study, seems to be the WHO histological grade, which however needs to be confirmed in a larger cohort of cases.

## Figures and Tables

**Figure 1 vetsci-09-00689-f001:**
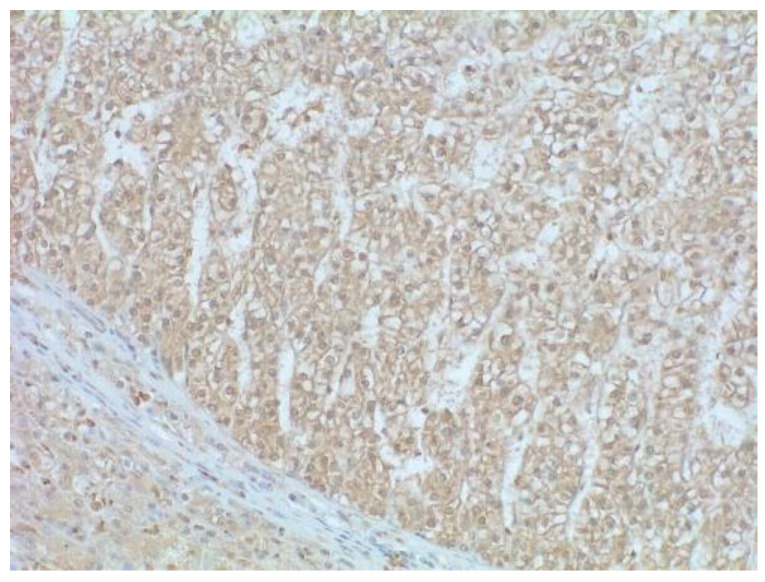
HCC well differentiated (WHO grade) trabecular pattern positive for PIVKA-II. In the image a portion of peritumoural hepatic tissue (bottom left) with a weak cytoplasmic positivity for PIVKA-II is also represented. Peritumoural fibrous capsule is negative.200× magnification–Mayer’s hematoxylin counterstaining.

**Figure 2 vetsci-09-00689-f002:**
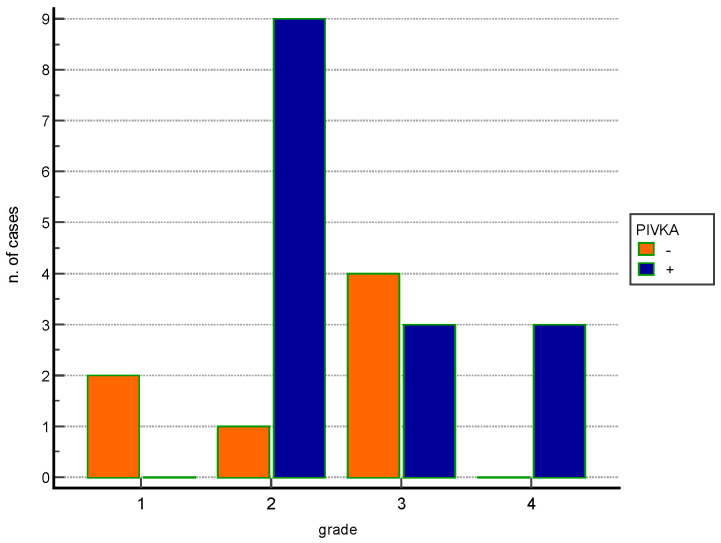
Histogram shows the distribution of positive and negative cases for PIVKA-II in relation to WHO histologic grade.

**Figure 3 vetsci-09-00689-f003:**
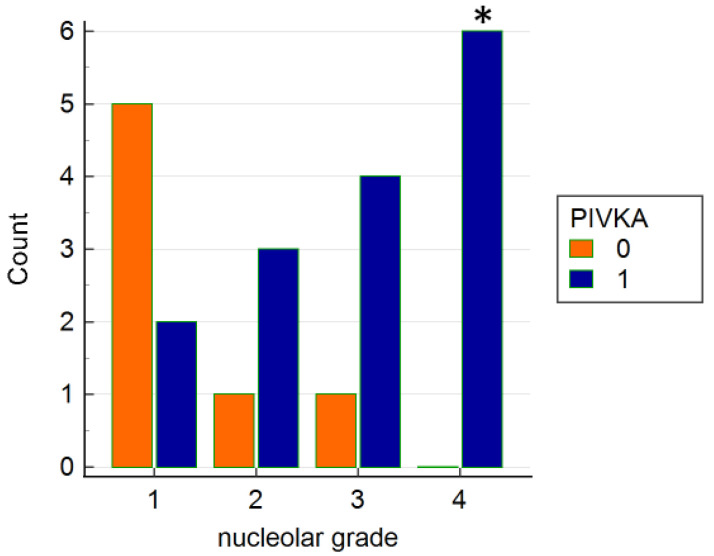
Histogram shows the distribution of positive and negative cases for PIVKA-II in relation to histologic grade relative to nucleolar pleomorphism. * = Fisher’s Exact test *p* < 0.05.

**Table 1 vetsci-09-00689-t001:** Histological grading system applied as proposed by Martins-Filho et al, 2017 [4].

**Nuclear Grade**	
I	Homogeneous, near-normal nuclei
II	Mild pleomorphism
III	Moderate pleomorphism, irregular distribution of chromatin
IV	Marked pleomorphism, bizarre nuclei
**Nucleolar Grade**	
I	Nucleoli barely seen at 400×
II	Evident nucleoli at 100–200×
III	Large nucleoli visible at 100×
IV	Prominent nucleoli visible at 40×
**Architectural Grade**	
I	Trabecular. 2–3 cell wide
II	Pseudoglandular pattern
III	Mild trabecular (4–10 cells wide)
IV	Macro-trabecular (<10 cells wide) or solid bizarre patterns

**Table 2 vetsci-09-00689-t002:** Signalment (breed, age, gender), histological grades, immunohistochemical result and follow up results. * censored case (alive after 2 years free from metastasis and/or local recurrences; dead for tumour unrelated reasons).

N°	Breed	Age	Sex	Nuclear Grade	Nucleolar Grade	Architecture Grade	WHO Grading	PIVKA	DFI Days	OS Days	Note
1	Samoyed	10	M	2	4	4	4	1	35	60	Metastases
2	Yorkshire	13	FS	1	2	2	2	1	730	730	*
3	Beagle	12	M	1	1	1	1	0	730	730	*
4	Mixed	10	FS	3	3	2	3	0	1	1	Death due to surgical complications
5	Golden R.	10	M	1	1	4	3	0	730	730	*
6	Dachshund	7	m	1	1	1	1	0	730	730	*
7	Mixed	13	Fs	1	1	4	3	0	730	730	*
8	Mixed	8	M	1	1	2	2	0	730	730	*
9	Mixed	9	FS	2	3	2	2	1	730	730	*
10	Mixed	13.5	M	2	3	2	2	1	730	730	*
11	Mixed	12	M	3	4	4	4	1	365	365	*
12	Cocker Spaniel	10	F	1	2	2	2	1	730	730	Relapsed at 900 days but alive
13	Mixed	11	FS	2	1	3	3	1	365	365	Metastases
14	Siberian Husky	13	FS	2	4	2	3	1	5	5	Death due to surgical complications
15	Cocker S.	11	FS	1	3	1	2	1	365	365	*
16	Golden R	8	M	1	3	3	2	1	730	730	*
17	Schnautzer	13	M	1	2	2	2	1	730	730	*
18	Golden R,	10	M	2	4	3	3	1	730	730	*
19	Mixed	12	FS	4	4	4	4	1	730	730	*
20	Rottweiler	12	MC	1	2	3	3	0	30	30	Metastases
21	Yorkshire	13	FS	2	4	1	2	1	730	730	*
22	Mixed	13	FS	2	1	1	2	1	730	730	*

## Data Availability

Not applicable.

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
