# Peer review of "Investigating a Prognostic Factor for Canine Hepatocellular Carcinoma: Analysis of Different Histological Grading Systems and the Role of PIVKA-II"

_vetsci, 2022, doi:10.3390/vetsci9120689_

Round 1

Reviewer 1 Report

This is a very interesting article, well written, with a clear description of the methods used. The figure legend could be improved, indicating the scale and the classification of the tumour (it seems well differentiated, but I would like confirmation). Is there any difference in the expression of PIVKA-II between well differentiated and undifferentiated tumours?

In the conclusion the authors state categorically that PIVKA-II is of no interest as a prognostic marker. This is a heavy statement for a study of 22 cases. Other studies with more cases may have different results. I would change the wording of the sentence a bit.

Overall, it is an excellent paper.

Author Response

We are very grateful to reviewer 1 for appreciating our work and for suggesting the minor revisions that were made and which improved the quality of our work.

1: The figure legend could be improved, indicating the scale and the classification of the tumour (it seems well differentiated, but I would like confirmation).

The figure legend has been updated as follows: "HCC well differentiated (WHO grade) trabecular pattern positive for PIVKA-II. In the image a portion of peritumoural hepatic tissue (bottom left) with a weak cytoplasmic positivity for PIVKA-II is also represented. Peritumoural fibrous capsule is negative.200x magnification – Mayer’s hematoxylin counterstaining "

Is there any difference in the expression of PIVKA-II between well differentiated and undifferentiated tumours?

In a first step, we recorded immunohistochemical positivity using a semi-quantitative method considering the percentage of positive cells and staining intensity. When positive, the neoplastic cells appeared diffusely positive and there was therefore only a difference in intensity between cases, but this was not statistically correlated with tumour differentiation. We thank you for this comment 

In the conclusion the authors state categorically that PIVKA-II is of no interest as a prognostic marker. This is a heavy statement for a study of 22 cases. Other studies with more cases may have different results. I would change the wording of the sentence a bit.

We are grateful for this valuable comment. The sentence has been updated as follows "In the present study, limited to the analyzed cohort of cases, PIVKA-II does not appear to be useful either as a diagnostic or prognostic marker in canine HCC differently to what has been observed in human HCC. This finding seems to be different to needs to be confirmed in a larger cohort of cases. The need remains to find reliable prognostic parameters in canine HCC that can predict its aggressive behaviour, which, although rarely observed, is present in some cases. The most promising method, limited to our study, seems to be the WHO histological grade, which however needs to be confirmed in a larger cohort of cases.

Lorella Maniscalco

 on behalf of the authors

Reviewer 2 Report

The article titled “Investigating a prognostic factor for canine hepatocellular carcinoma: analysis of different histological grading systems and the role of PIVKA-II” submitted to veterinary science manuscript number vetsci-2071932.

In this manuscript, the authors have investigated have The relationship between PIVKA-II expression and prognosis in hepatocellular carcinoma, which is relatively rare in dogs, was investigated histopathologically using 22 cases. Although this study is based on a small number of cases (22), it is a highly valuable study aimed at establishing an effective prognostic marker for hepatocellular carcinoma, which is rarely analyzed in veterinary medicine.

              I feel the content of this paper is consistent with the intent of the special issue " Comparative Pathology of Cancers in Animals". However, I feel that the details of this paper are incomplete, and we minor comment below to improve the manuscript.

 Paragraphs 2 and 3 of the introduction section (lines 46-52) should be combined into one paragraph since they both cite the latest findings.

 Regarding fig3, the author says that there is a statistical association between nucleolar pleomorphism grade 3 and negativity to PIVKA-II, but the statistically significance (e.g. p value) is not shown in the figure. The author should clearly state the statistically significant difference.

 Line 220 “rognostic” may be mistyped “prognostic”

Author Response

We are very grateful to reviewer 2 for appreciating our work and for suggesting the minor revisions that were made and which improved the quality of our work.

We respond below in bold to each consideration 

 Paragraphs 2 and 3 of the introduction section (lines 46-52) should be combined into one paragraph since they both cite the latest findings.

We agree. The paragraphs have been combined as suggested

 Regarding fig3, the author says that there is a statistical association between nucleolar pleomorphism grade 3 and negativity to PIVKA-II, but the statistically significance (e.g. p value) is not shown in the figure. The author should clearly state the statistically significant difference.

We are very grateful for this comment. The statistical significant result was between grade 4 nucleolar pleomorphism and positivity to PIVKA-II. It is possible that an accident happened during the review process between the authors. Now has been stated as follows and images has been modified with an asterisk to indicate significance: "

Moreover analyzing the others histological grades a statistical association between nucleolar pleomorphism grade 4 and positivity to PIVKA-II (Fisher’s Exact test P<0.05) was found (Figure 3).

 Line 220 “rognostic” may be mistyped “prognostic”

corrected as suggested

Lorella Maniscalco 

on behalf of the authors